# An Acidic Polysaccharide with Anti-Inflammatory Effects from Blackened Jujube: Conformation and Rheological Properties

**DOI:** 10.3390/foods11162488

**Published:** 2022-08-17

**Authors:** Chuang Liu, Fangzhou Wang, Rentang Zhang

**Affiliations:** Key Laboratory of Food Processing Technology and Quality Control of Shandong Higher Education Institutes, College of Food Science and Engineering, Shandong Agricultural University, 61 Daizong Street, Tai’an 271018, China

**Keywords:** blackened jujube, polysaccharides, conformation, rheological properties, anti-inflammatory effects

## Abstract

An acidic polysaccharide fraction (BJP-4) was isolated from blackened jujube, and its advanced structures and anti-inflammatory activity were investigated. X-ray diffraction showed that BJP-4 exhibits both crystalline and amorphous portions. Atomic force microscopy data suggested that it contains a large number of spherical lumps. Circular dichroism and Congo red experiments revealed that it has no triple-helix conformation. In steady shear flow results, the BJP-4 solution was a pseudoplastic non-Newtonian fluid with acid-base stability. BJP-4 (20 mg/mL) showed liquid-like properties (G″ > G′), while it performed weak gel-like behavior at a high concentration (40 mg/mL) (G′ > G″). The anti-inflammatory effects of BJP-4 were further evaluated through in vitro experiments. BJP-4 could down-regulate the over-secretion of inflammatory factors (NO, IL-6, IL-1β, TNF-α, iNOS and COX-2) in RAW264.7 cells due to LPS stimulation. Moreover, it demonstrated that BJP-4 restrained the NF-κB signal pathway by regulating TLR4 expression, reducing IκBα phosphorylation level and NF-κB p65 nuclear translocation. In summary, this present study contributes to the application of blackened jujube polysaccharides in the foods and medicine field.

## 1. Introduction

Inflammation is the body’s initial response to the stimulation of invasive harmful substances such as bacteria, viruses, and other pathogens [1]. It is generally believed that inflammation is the basic pathological process of disease and is associated with the pathogenesis and progress of various diseases [2,3]. Inflammation-related factors, including interleukin 6 (IL-6), tumor necrosis factor-alpha (TNF-α), interleukin 1 beta (IL-1β), cyclooxygenase-2 (COX-2), and inducible nitric oxide synthase (iNOS) play a critical role as promoters and indicators [4]. In these mediators, TNF-α, IL-6, and IL-1β are crucial cytokines for pro-inflammation, which can directly express the level of inflammation. The iNOS can promote nitric oxide (NO) secretion in cells, thereby stimulating the expression of various proteins and enzymes that play a key role in inflammation [5]. In addition, COX-2 is an inducible synthase, which is the key enzyme in initiating inflammatory responses [6]. Therefore, modulating inflammatory factors is one of the effective pathways to suppress the inflammatory response.

RAW264.7 macrophages, as the major members of the body’s immune function, perform vital physiological actions during inflammation and are greatly used for cell biology research [7,8]. Studies have confirmed that when Toll-like receptor 4 (TLR4) received exogenous stimulation, the inhibitor kappa B alpha (IκBα) was phosphorylated and degraded, and then dissociated from nuclear factor kappa B (NF-κB) [9]. Then, the NF-κB p65 subunit enters the nucleus after being activated, further initiating the expression of related enzymes (COX-2 and iNOS) and the production of pro-inflammatory factors such as IL-6, IL-1β, and TNF-α [10]. Therefore, the TLR4/NF-κB signaling pathway is consistently recognized by most researchers as a critical pathway for treating inflammation.

Jujube, broadly cultivated in China, has many bioactive compounds such as polysaccharides, polyphenols, vitamins, and adenosine cyclophosphate, of which polysaccharides account for more than 60% of dry weight [11]. Jujube polysaccharides were reported to exhibit various pharmacological effects, including antioxidant, anti-inflammatory, anti-tumor, and hepatoprotective activities [12]. Notably, a polysaccharide from *Z. jujuba cv. Junzao*, rich in galacturonic acid and with a molecular weight (MW) of 153.3 kDa, exhibited significant anti-inflammatory effects by inhibiting the secretion of TNF-α, NO, and COX-2 in RAW264.7 macrophages [13]. In addition, it was reported that a polysaccharide from the fruit of *Z. jujuba cv. Muzao*, which mainly contained arabinose (4.52%) and galactose (2.64%) with an MW of 89.90 kDa, inhibited the levels of IL-6 and TNF-α in RAW264.7 cells [14]. Blackened jujube is an elaborate food made of dried jujube fermented at 70–75 °C under a certain humidity for 5–7 days [15,16]. The in vitro antioxidant capacity of blackened jujube polysaccharides (BJPs) was reported, confirming the free radical scavenging activity and total reduction power, and exhibiting therapeutic effects on H_2_O_2_-stimulated HUVECs [17]. Considering the close relationship between anti-inflammatory capacity and oxidation imbalance, we conjecture that BJPs have excellent anti-inflammatory activity.

The biological functions of polysaccharides not only change with the primary structure (MW, monosaccharide composition, and sugar chain sequence) but are also closely related to the conformation [18,19,20]. For example, it is generally believed that polysaccharides with triple helix structures exhibit more excellent biological activities [21]. The structure results of jujube polysaccharides are quite different, mainly affected by jujube varieties, the extraction technology, purification process, and detection method. The advanced structure and anti-inflammatory effects of BJPs have been poorly reported; thus, exploring their structure-effect relationships was of great significance.

Following the previous method, BJP-4 was extracted using ultrasonic-assisted hot water and purified by a DEAE-cellulose 52 anion-exchange column and Sephadex G-100 gel column. In addition, BJP-4 had a high yield (12.36%) and excellent antioxidant activity with an IC_50_ of 0.54 mg/mL and 0.58 mg/mL for DPPH and ABTS+ radical scavenging [17], respectively. As so, this fractionation was selected for further study. In this study, the conformational analysis of BJP-4 is systematically estimated by X-ray diffraction (XRD), atomic force microscopy (AFM), circular dichroism (CD), and Congo red experiments. Then, the anti-inflammatory effect of BJP-4 is evaluated with Lipopolysaccharide (LPS)-treated RAW264.7 cells as the model. These results could provide the theoretical link between the advanced structure and anti-inflammatory activity of blackened jujube polysaccharide and demonstrate its potential value for development as a novel anti-inflammatory drug. The effects of mass concentration and pH conditions on the rheological properties of blackened jujube polysaccharide are also investigated to provide guidance for its application in the food field.

## 2. Materials and Methods

### 2.1. Materials

*Z. jujuba cv. Hamidazao* fruits were provided by Guorentang Food Technology Co., Ltd. (Shandong, China). RAW264.7 macrophages were obtained from Wanlei Biotechnology Co., Ltd. (Shenyang, China). High glucose Dulbecco’s modified eagle medium (DMEM), fetal bovine serum (FBS), and penicillin were provided from Gibco Biotechnology Co., Ltd. (Grand Island, NY, USA). LPS was provided by Sigma Chemical Co. (St. Louis, MO, USA). Kits for TNF-α, NO, IL-1β, cell counting kit 8 (CCK-8), and IL-6 were supplied by Shanghai Enzyme-linked Biotechnology Co., Ltd. (Shanghai, China). The Bicinchoninic Acid (BCA) protein kit and protein extraction kit were purchased from Beyotime Biotechnology Co., Ltd. (Shanghai, China). The antibodies against TLR4, IKBα, NF-κB p65, p-IKBα, p-NF-κB p65, COX-2, and iNOS were obtained from Boster Biological Technology (Wuhan, China). The primers of TNF-α, IL-1β, IL-6, COX-2, and iNOS were bought from Sangon Biotech Co. Ltd. (Shanghai, China). The total RNA isolation kit was supplied by BioTeke Corporation (Beijing, China). BeyoRT II M-MLV Reverse Transcriptase was purchased from Beyotime Biotechnology Co., Ltd. (Shanghai, China). SYBR Green Master Mix was supplied from Solarbio Science & Technology Co., Ltd. (Beijing, China). Congo red was supplied from Tianjin Zhiyuan Chemical Reagent Co., Ltd. (Tianjin, China). All other chemicals and reagents were of analytical reagent grade.

### 2.2. Molecular Morphology and Conformational Analysis of BIP-4

#### 2.2.1. XRD Analysis

Crystallization of BJP-4 was measured using XRD (D8 Advance, Bruker, Germany). Briefly, 2 mg of BJP-4 was weighed and placed on the sample plate, and carefully flattened. The operating conditions were applied as follows: Cu Kα radiation, 40 kV, 40 mA, 2θ = 5° − 80°, and a speed of 3°/min [22].

#### 2.2.2. AFM Measurement

BJP-4 samples were prepared at a concentration of 1 μg/mL then transferred to the mica carrier and dried at 25 °C. The AFM (BioScope Catalyst NanoScope V, Bruker, Billerica, MA, USA) was used to tap the dry mica flake and present images [23].

#### 2.2.3. CD Analysis

A J-815 Chirascan spectropolarimeter (JASCO Corp, Tokyo, Japan) was adopted to analyze BJP-4 (2 mg/mL). The scanning wavelength coverage was 185–400 nm using a 1 mm quartz cell [24].

#### 2.2.4. Congo Red Experiment

The Congo red experiment was performed as previously reported [25]. Briefly, 1.0 mL of 2.0 mg/mL of BJP-4 solutions were added with 1.5 mL of Congo red solution (0.2 M), followed by NaOH solution (0–0.8 M) to give a final concentration of NaOH (0, 0.5 M, 0.1 M, 0.15 M, 0.2 M, 0.3 M, 0.4 M). The reaction solution was detected with a UV2450 spectrophotometer (Shimadzu, Kyoto, Japan) in the range of 400 to 800 nm.

### 2.3. Rheological Measurements

The rheological characterization was performed on a rheometer (MCR 102, Anton Paar, Germany) at 25 °C with a parallel board (diameter 50 mm, clearance 1 mm).

#### 2.3.1. Steady Shear Analysis

##### Effect of Concentration on Viscosity Characteristics of BJP-4

A series of BJP-4 sample solutions with concentrations of 2.5, 5.0, 10, 20, 40 mg/mL in deionized water was prepared. Sample flow curves were measured in viscosity and flow curve templates, where the shear rate range was set to 0.1–100 s^−1^.

##### Viscosity Properties of BJP-4 at Different pH

The pH values of 20 mg/mL BJP-4 solution were adjusted to around 3.0, 7.0, and 11 using 1.0 M NaOH and HCl. The equipped solutions were measured as mentioned in the Effect of Concentration on Viscosity Characteristics of BJP-4.

#### 2.3.2. Dynamic Oscillatory Analysis

Two types of BJP-4 solution (20 mg/mL and 40 mg/mL) were obtained as detailed in the Effect of Concentration on Viscosity Characteristics of BJP-4. Before dynamical oscillatory frequency measurement, the sample strain value was obtained using the linear viscoelastic region mode with a scanning range of 0.1–100%. According to the strain value, oscillatory measurements were implemented in the frequency range of 10^−2^–10^1^ Hz at 25 °C and then the elastic modulus (G′) and viscous modulus (G″) of the polysaccharide solutions were recorded. The loss factor (tanδ) was derived from the following Equation (1).
tanδ = (G″/G′) × 100%(1)

### 2.4. Anti-Inflammatory Effect of BJP-4 In Vitro

#### 2.4.1. Cell Culture and Viability

A DMEM medium containing penicillin (100 U/mL) and 10% FBS was used to incubate RAW264.7 macrophages at 37 °C with 5% CO_2_.

Assessment of the cytotoxicity of BJP-4 in a certain concentration range (12.5–200 μg/mL) was performed using the CCK-8 method. Briefly, cells were cultured to the logarithmic growth phase and transferred to 96-well plates at a concentration of 3 × 10^3^ cells/well, and then treated with 1 μg/mL LPS and the corresponding concentration of BJP-4. After 24 h of treatment in an incubator at 37 °C with 5% CO_2_, CCK-8 was added and incubated for 2 h. The microplate reader (ELX-800, Biotek, Winooski, VT, USA) was applied to determine the optical density (OD) of the solution at 450 nm. Cell viability was computed as follows:Cell viability = (ODsample/ODblank) × 100%(2)

#### 2.4.2. Measurement of NO and Cytokines

Cells culture was as described in Section 2.4.1. Cell culture supernatant was aspirated and cytokine expressions were detected using ELISA kits based on the instructions.

#### 2.4.3. Quantitative Real-Time Polymerase Chain Reaction (qRT-PCR)

The gene expression levels in the RAW264.7 macrophages were determined by qRT-PCR, as previously reported [26]. TRIpure reagent was utilized to extract total cellular RNA, and then a spectrophotometer (Nanodrop 2000c, Thermo Fisher Scientific, Waltham, MA, USA) was applied to detect RNA concentration. The cDNA was synthesized from total RNA through Transcriptor Reverse Transcriptase. The amplification of the qRT-PCR used SYBR Green Master Mix on a system Exicycler 96 (Bioneer, Daejeon, South Korea). The amplification conditions were 94 °C for 5 min, 60 °C for 20 s (40 cycles), 72 °C for 150 s, and 40 °C for 90 s. The specific primers’ sequences are listed in Table 1. The experimental analysis was performed using the 2^−ΔΔCt^ method.

#### 2.4.4. Western Blot

The cell treatments were the same as in Section 2.4.1. The Western blot analysis was based on the method in the literature [27]. Briefly, the proteins were isolated and quantified using protein extraction and BCA protein assay kits, respectively. Various concentrations of polyacrylamide gels (5–14%) were prepared and subjected to protein electrophoresis, after which the proteins were transferred to polyvinylidene difluoride (PVDF) film. PVDF films were removed and immersed in Tris-buffered saline with Tween 20 (TBST) and shaken for 5 min, then the membranes were immersed in 5% (*m*/*v*) skim milk powder solution and shaken for 1 h. Conditions of the primary antibody incubation were TLR4 (1:500), IκBα (1:1000), NF-κB p65 (1:1000), p-IκBα (1:1000), p-NF-κB p65 (1:1000), iNOS (1:500), COX-2 (1:1000), β-actin (1:1000), and Histone H3 (1:1000), overnight at 4 °C. After that, the films were removed from the hybridization bag and soaked in TBST for four washes, and then the goat anti-rabbit IgG HRP secondary antibodies were incubated at a dilution ratio of 1:5000 for 45 min at 37 °C. After washing six times, the ECL reagent was added to the membranes, which were photographed by the WD-9413B imaging system (Beijing LIUYI Biotechnology Co., Ltd., Beijing, China).

### 2.5. Statistical Analysis

Results are denoted as mean ± standard deviation (SD). The significance of differences was evaluated using a one-way analysis of variance (ANOVA) followed by Tukey’s test using SPSS software with *p* < 0.05.

## 3. Results and Discussion

### 3.1. Morphological Properties of BJP-4

XRD can provide key information about the structure of biomaterials, such as the crystalline or amorphous properties of polysaccharides. As shown in Figure 1A, BJP-4 had a “bun-shaped” peak in the range of 10 to 15°. In addition, there were two sharp and narrow characteristic peaks that appeared at 21.1° and 22.4°, corresponding to the crystalline nature of BJP-4 [28]. Meanwhile, there were small diffraction absorption peaks in the 40–50° range related to the uronic acid found in acidic polysaccharides [29]. In summary, it indicated that BJP-4 exists in both a partially crystalline structure and amorphous fraction. This result was the same as previously reported for *Althaea officinalis* L. root [30]. AFM is commonly applied to observe the morphological characteristics of biological substances. As shown in Figure 2B,C, there were numerous spherical lumps of BJP-4, suggesting molecular aggregation. These can be explained by the interaction of carboxyl and hydroxyl groups of polysaccharides and the formation of molecular interactions with water molecules, which cannot be ignored [31]. The morphology of BJP-4 was different from that of *Muzao* [32], *Ruoqiangzao* [29], and *Linzexiaozao* [11], which might be related to galacturonic acid and hydrogen [28].

### 3.2. Advanced Structure of the Polysaccharide BJP-4

CD is considered to be a classical method for analyzing the conformational variation in polysaccharide secondary structures. The CD spectra of BJP-4 was shown in Figure 1D. No negative absorption peak was found over the 185–400 nm wavelength range, indicating that BJP-4 contained no helical structure. However, a large positive absorption spike was recorded at 190 nm, which proved to be an asymmetric structure in BJP-4. This was primarily caused by the folding, flipping, winding, and irregular morphology of polysaccharide molecules in an aqueous solution [33]. Congo red analysis was conducted to identify whether a triple helix structure was involved in BJP-4. The maximum absorption (λ_max_) changes of BJP-4 and Congo red complex were shown in Figure 1E. No red shift in λ_max_ of the complex was observed in Congo red solution as blank control, indicating that there was no triple helix structure in BJP-4, which was consistent with the results of CD. This phenomenon was in accordance with the previous research [30]. Numerous studies have shown that the triple helix structure and MW of polysaccharides significantly affect their pharmacological activity. For example, under the condition of similar molecular weight, the triple helix structure had better antitumor effects than the single helix structure of lentinan [34]. However, the structure-function relationships of polysaccharides are difficult to reveal systematically and more studies are needed to provide theoretical references.

### 3.3. Rheological Measurements

#### 3.3.1. Steady Shear Flow Properties

The apparent viscosities of the BJP-4 solutions at different concentrations (2.5–40 mg/mL) are displayed in Figure 2A. From the 0.1–100 s^−1^ shear rate range, the apparent viscosity of BJP-4 decreased with increasing shear rate and exhibited shear thinning behavior, which was a typical pseudoplastic non-Newtonian fluid. This behavior was mainly due to the change of the link point between molecules as the shear rate increased, which made the resistance between molecules smaller and led to a decrease in viscosity. The apparent viscosity of BJP-4 showed a significant concentration-dependent effect, where the viscosity of 40 mg/mL BJP-4 could reach 2000 mPa·s at a shear rate of 0.1 s^−1^. This phenomenon was explained by the increasing number of polysaccharide molecules per unit volume and the enhancement of molecular association, resulting in an increase in the degree of polymerization and an increase in the flow resistance of the solution. Similar shear-thinning characteristics have been reported for polysaccharides extracted from *Basil seed gum*, *Dendrobium officinale*, and *Extreme salt-tolerant Bacillus subtilis* LR-1 [35,36,37].

Figure 2B shows the viscosity characteristics of BJP-4 (20 mg/mL) samples with different pH values ranging from 0.1–100 s^−1^. The viscosity of BJP-4 at pH = 3.0 and pH = 11 was less than that at pH = 7.0, which was attributed to the hydrogen bond cleavage of BJP-4 polysaccharide in an acidic or alkaline environment, thus promoting the decomposition of polysaccharide and reducing its apparent viscosity. However, the comprehensive results showed that the BJP-4 polysaccharide solution has good acid-base stability, so it could be used in acidic and alkaline foods.

#### 3.3.2. Dynamic Oscillation Shear Test

The linear viscoelastic region was measured for different concentrations (40 mg/mL and 20 mg/mL) of BJP-4 solution showing results of 1–10% and 30–100%, respectively. In this experiment, frequency oscillation tests were carried out under 5% and 50% strain values, respectively. As can be seen in Figure 2C–F, the modulus G″ (loss modulus) was less than the modulus G′ (storage modulus) at a concentration of 40 mg/mL and exhibited a strong elastic gel-like behavior with tanδ < 1 at all frequencies. On the contrary, 20 mg/mL of BJP-4 showed viscous behavior. This phenomenon was similar to Yam polysaccharide, which exhibited “liquid-like” behavior below 14% (*w*/*v*) and “gel-like” behavior for 14–20% (*w*/*v*) [38].

### 3.4. Anti-Inflammatory Effects of BJP-4 In Vitro

#### 3.4.1. Evaluation of BJP-4 on Cell Viability and NO Production

The assessment of the cytotoxicity of BJP-4 on RAW264.7 cells used a CCK-8 assay. As presented in Figure 3A, no significant cytotoxicity of BJP-4 was observed below 50 μg/mL. However, 100 μg/mL of BJP-4 treatment caused a 22.7 ± 9.2% reduction in cell viability compared to the blank control (*p* < 0.05). In RAW264.7 cells, excessive NO production can lead to a systemic inflammatory response [39,40]. As can be seen in Figure 3B, the NO secretion of RAW264.7 cells after induction by LPS increased by 13.19 ± 1.0 μM, which was extremely significantly higher than that of the blank control group (*p* < 0.01), showing that the inflammation model was successfully established. It was noteworthy that BJP-4-treated groups showed better inhibition of NO secretion only at doses higher than 25 μg/mL (*p* < 0.05), and the anti-inflammatory effect of BJP-4 was concentration-dependent over a range of dose concentrations (25−200 μg/mL). In particular, under the stimulation of 25 and 50 μg/mL BJP-4 secreted 13.9 ± 1.5 and 11.3 ± 1.5 μM of NO, respectively, which was 18.2 ± 8% and 33.5 ± 8% less than those of the positive control (LPS). Combining the above results, three optimal concentrations of BJP-4 (12.5, 25, 50 μg/mL) were chosen for follow-up tests.

#### 3.4.2. Effects of BJP-4 on the Mediation of Pro-Inflammatory Cytokines

LPS is known to cause an inflammatory response in RAW264.7 macrophages, resulting in the over-secretion of pro-inflammatory cytokines represented by IL-6, IL-1β, and TNF-α [41]. As exhibited in Figure 3C–E, after stimulation of RAW264.7 cells with LPS, the levels of TNF-α, IL-1β, and IL-6 increased from 42.45, 35.67 and 25.51 pg/mL to 220.40, 172.16, and 97.27 pg/mL, respectively (*p* < 0.01). In contrast, supplementation with different doses of BJP-4 (12.5−50 μg/mL) significantly reversed the trend of increased expression in pro-inflammatory factors compared to the LPS group (*p* < 0.01), and in a concentration-dependent manner. Especially, TNF-α, IL-1β and IL-6 contents were reduced by 57.94%, 42.26%, and 12.26%, in the high dose of BJP-4 (50 μg/mL) treatment, respectively, with the LPS group as the control (*p* < 0.01). BJP-4 exhibited excellent suppression of LPS-triggered inflammatory reactions in RAW264.7 macrophages, demonstrating the capacity of BJP-4 to exert anti-inflammatory effects. Polysaccharides from *purple sweet potato*, *Arctium lappa* and *Sea cucumber Stichopus chloronotus* could ameliorate inflammation in RAW264.7 cells via the rebalancing of inflammatory cytokines [42,43,44].

#### 3.4.3. Regulation of BJP-4 on mRNA Expression of Multiple Cytokines

The effects of stimulation of RAW264.7 cells with different products (LPS and BJP-4) on their mRNA expression levels of pro-inflammatory cytokines (IL-6, TNF-α, IL-1β, COX-2, and iNOS) were further examined by qRT-PCR. As seen in Figure 4A–C, LPS treatment alone significantly increased TNF-α production by 5.09-fold, IL-1β production by 5.19-fold, and IL-6 production by 4.08-fold, respectively, in comparison with the blank control (*p* < 0.01). BJP-4 was found to interfere with the mRNA expression of pro-inflammatory cytokines in a similar fashion to its impacts on secretion levels, exhibiting a dose-dependent down-regulation of gene expression. Notably, the gene expressions of TNF-α, IL-1β, and IL-6 by BJP-4 (25 μg/mL) treatment were reduced to 38.70%, 44.30%, and 31.37% of the LPS group, respectively (*p* < 0.01), and suppressed 68.96%, 63.58%, and 57.35% by BJP-4 (50 μg/mL), respectively (*p* < 0.01). The iNOS, a synthase that promotes NO production, is secreted by macrophages after inflammation. COX-2, an inducer enzyme, is expressed by macrophages subjected to multiple stimuli (e.g., pro-inflammatory factors). As can be seen from Figure 4D,E, the addition of LPS resulted in overexpression of iNOS and COX-2 gene levels in RAW264.7 macrophages, which increased by 4.05- and 4.31-fold, respectively, using the blank control group as a reference (*p* < 0.01). The inhibitory effect of three concentrations of BJP-4 on cytokine gene expression was enhanced as the dose increased (*p* < 0.01), especially when it was 50 μg/mL; the iNOS and COX-2 mRNA expressions accounted for 37.78% and 38.05% of the LPS group, respectively (*p* < 0.01). Additionally, similar trends were observed for the protein expression of iNOS and COX-2. Similar results were also observed in RAW264.7 macrophages with LPS-triggered inflammation in response to the supplementation of *Moringa oleifera* roots, *Umbilicaria yunnana*, and *Rubus chingii Hu* polysaccharides [45,46,47].

#### 3.4.4. BJP-4 Inhibited the TLR4/NF-κB Pathway

To clarify whether BJP-4 exhibited anti-inflammatory ability via the TLR4/NF-κB pathway, the Western blot method was used to measure the related protein expressions (TLR4, NF-κB p65, IκBα, p-NF-κB p65, p-IκBα) (Figure 5). The phosphorylation levels of NF-κB p65 and IκBα are important indicators to confirm the initiation of the NF-κB pathway. In addition, TLR4 on the outer cell membrane binds to LPS on the cell wall to activate the intracellular NF-κB signaling cascade. As exhibited in Figure 6, LPS treatment elevated the protein expression levels of TLR4, p-NF-κB p65/NF-κB p65 and p-IκBα/IκBα by 2.54-, 7.6-, and 33.52-fold, respectively, among the blank controls (*p* < 0.01). However, the administration of BJP-4 dose dependently reversed this activation, especially the TLR4, p-NF-κB p65/NF-κB p65, and p-IκBα/IκBα levels from BJP-4 (50 μg/mL) treatment were reduced to 45.48%, 13.95%, and 4.07% of the LPS group, respectively (*p* < 0.01). As shown in Figure 5F, the nuclear NF-κB p65 was initiated by the LPS group (10.51-fold of blank control), while BJP-4 significantly decreased NF-κB content in a dosage-dependent fashion (*p* < 0.01). To summarize, these analyses demonstrated that BJP-4 could mediate the inflammatory process by suppressing the LPS-treated secretion of pro-inflammatory cytokines in RAW264.7 macrophages through the TLR4/NF-κB signaling pathway. Consistent with these findings, suppression of TLR4, IκBα, and NF-κB activation was also observed in LPS-stimulated RAW264.7 cells treated with other polysaccharides [48,49].

## 4. Conclusions

In the present study, a blackened jujube polysaccharide BJP-4 was found to have both semi-crystalline and amorphous portions and was identified to have no triple-helix conformation. In addition, BJP-4 exhibited non-Newtonian shear-thinning flow behavior and had good acid-base stability at pH 3.0–11. The dynamic shear rheological properties showed that BJP-4 exhibited weak gel behavior (G″ < G′) at a high concentration (40 mg/mL). BJP-4 is a non-gel polysaccharide and could be used as a food thickener. Further analysis found that BJP-4 had an excellent anti-inflammatory effect against RAW264.7 cells, which could obviously restrain the secretion of NO, IL-6, TNF-α, IL-1β, COX-2, and iNOS. Moreover, BJP-4 remarkably down-regulated the protein expression of TLR4, p-IκBα/IκBα, and p-NF-κB p65/NF-κB p65, suggesting that the TLR4/NF-κB signaling pathway performs a crucial action in the anti-inflammatory effect of BJP-4. This research could contribute to theoretical support for the conformation-activity relationship of blackened jujube polysaccharides, which is of vital relevance to the exploration of novel foods and anti-inflammatory drugs. Because of the complex structure of polysaccharides, the anti-inflammatory activity for a single fraction of blackened jujube polysaccharides and whether its structure–activity relationship is relatively limited remains to be revealed and requires further systematic study. Moreover, the anti-inflammatory activity of polysaccharides and their mechanisms of action are more investigated at the cellular and animal levels, and there is a need for further clinical trial studies.

## Figures and Tables

**Figure 1 foods-11-02488-f001:**
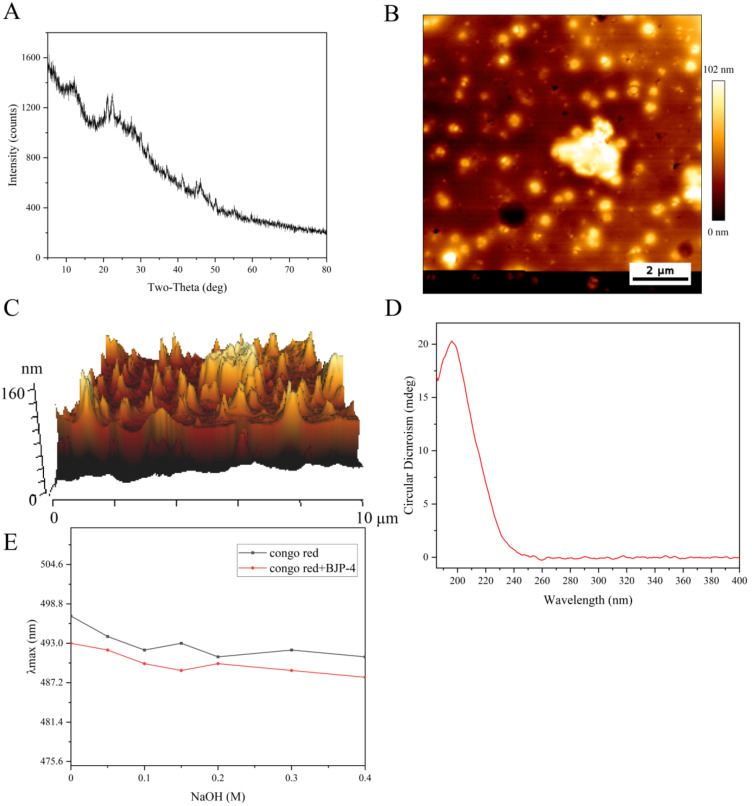
The chemical structure analysis of BJP-4: (**A**) X-ray diffraction pattern. (**B**) The atomic force microscope planar image. (**C**) Three-dimensional image of atomic force microscope. (**D**) Circular dichroism spectrum. (**E**) Effect of BJP-4 on the λmax of Congo red.

**Figure 2 foods-11-02488-f002:**
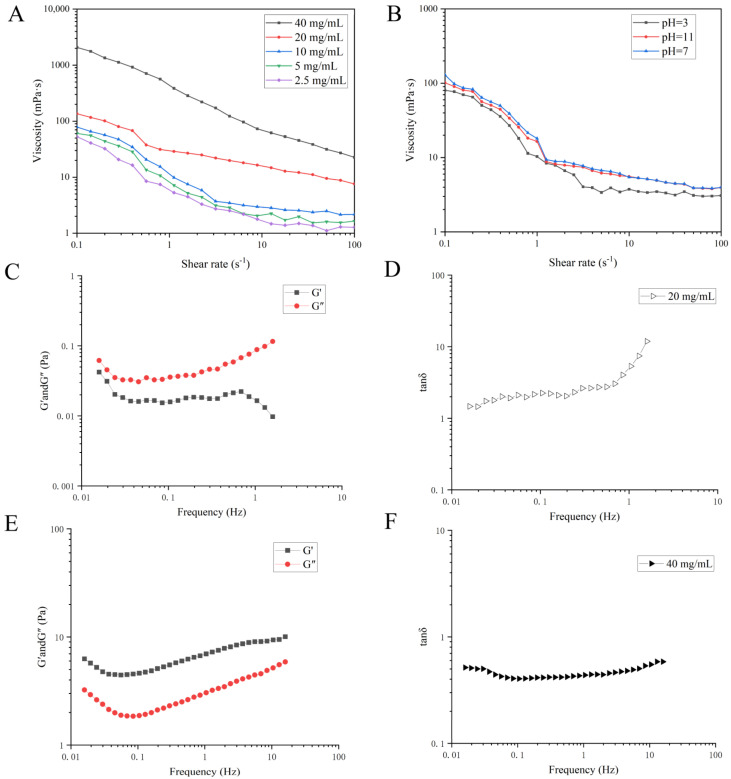
The rheological measurements of BJP-4: (**A**) Variation in apparent viscosity of different concentration BJP-4 with shear rate. (**B**) Variation in apparent viscosity of 20 mg/mL BJP-4 with pH change. (**C**) Frequency sweeps performance at 50% strain for determining the storage modulus G′ and loss modulus G′ for 20 mg/mL BJP-4 solution. (**D**) The tanδ of 20 mg/mL BJP-4. (**E**) Frequency sweeps performance at 5% strain for determining the storage modulus G′ and loss modulus G″ for 40 mg/mL BJP-4 solution. (**F**) The tanδ of 40 mg/mL BJP-4.

**Figure 3 foods-11-02488-f003:**
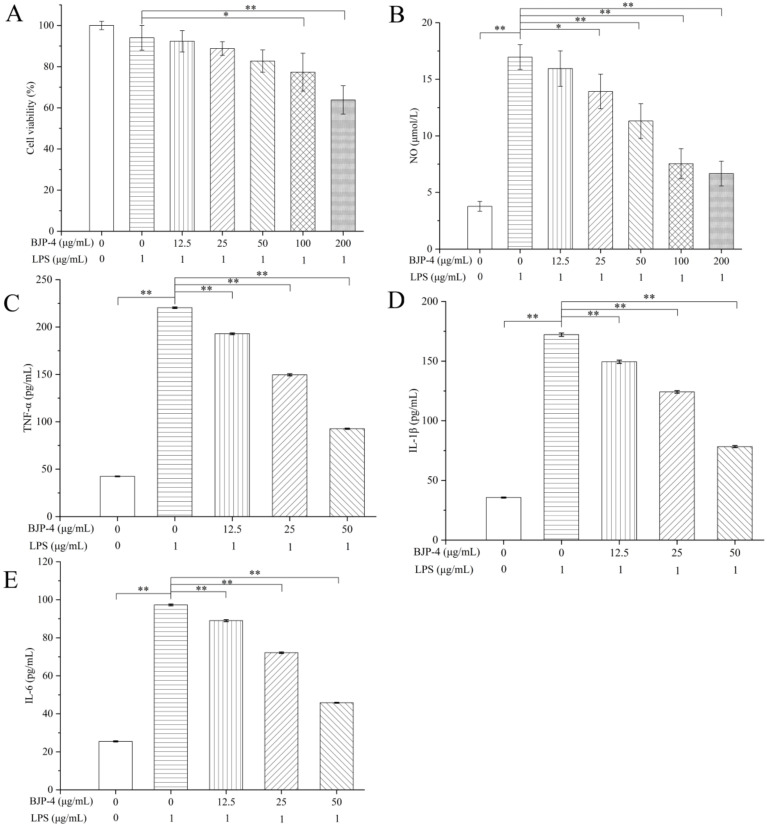
Comprehensive evaluation of the anti-inflammatory activity of BJP-4: (**A**) Effect of BJP-4 on the viability of RAW264.7 cells. (**B**) Inhibition of BJP-4 on NO, (**C**) TNF-α, (**D**) IL-1β, (**E**) IL-6 production induced by LPS in RAW264.7 macrophages. Values are expressed as mean ± SD. * *p* < 0.05; ** *p* < 0.01.

**Figure 4 foods-11-02488-f004:**
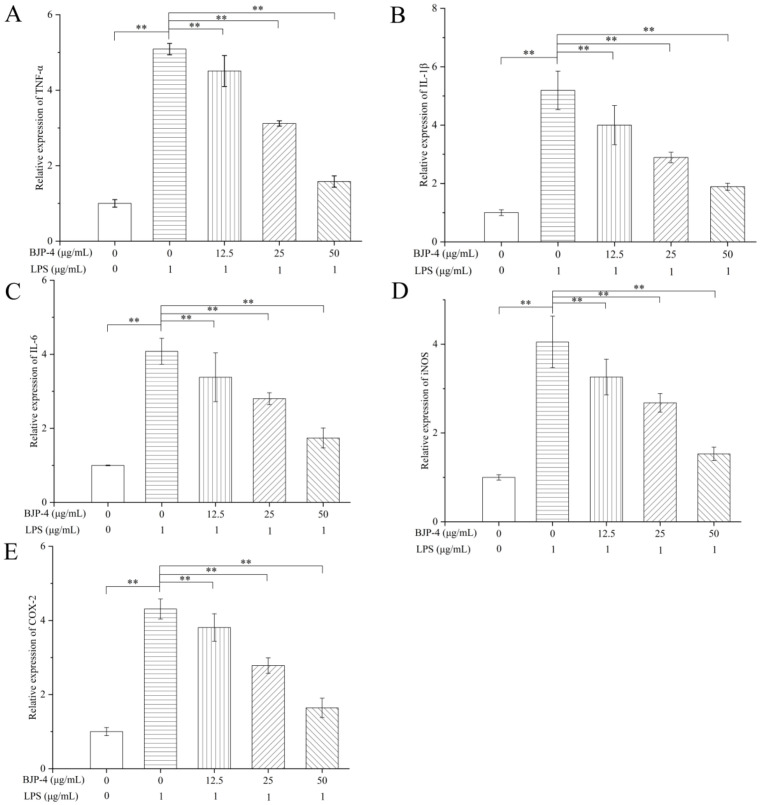
Effects of BJP-4 on LPS-stimulated mRNA expressions: (**A**) The mRNA levels of TNF-α, (**B**) IL-1β, (**C**) IL-6, (**D**) iNOS, (**E**) COX-2. Values are expressed as mean ± SD. ** *p* < 0.01.

**Figure 5 foods-11-02488-f005:**
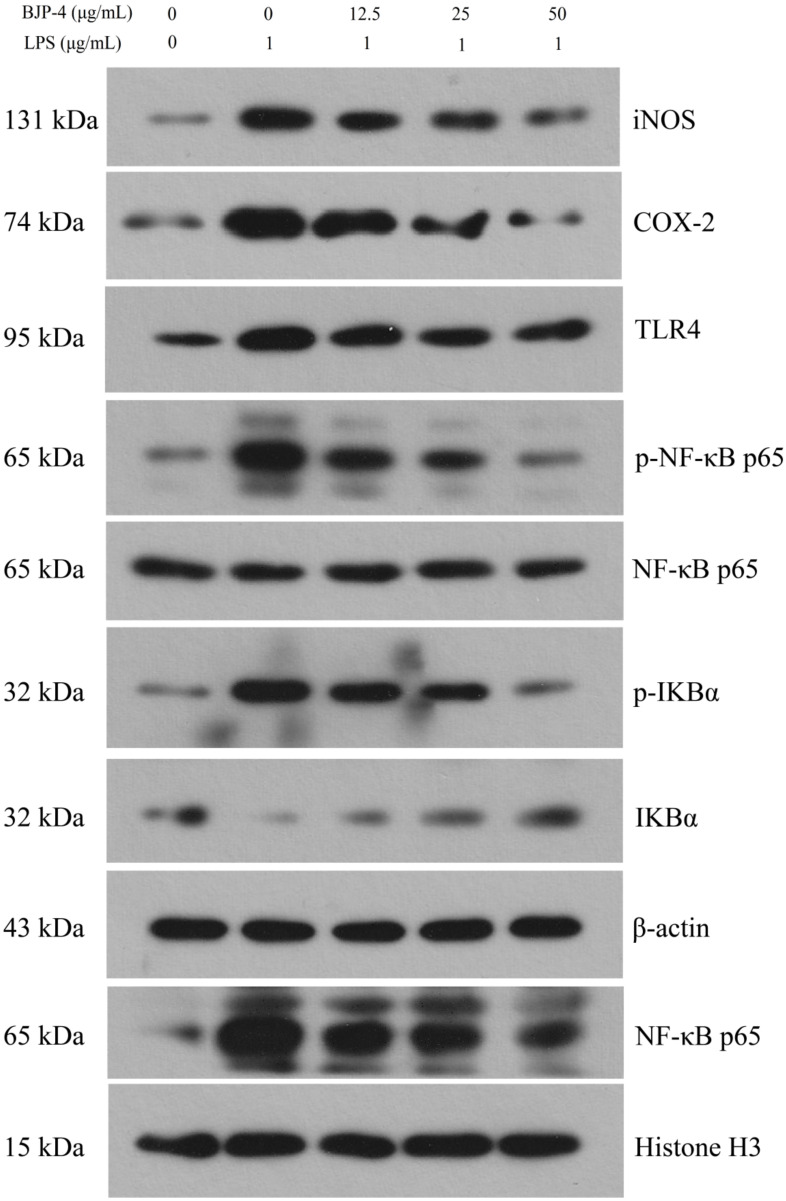
Western blot analysis of BJP-4 on the TLR4/NF-κB signaling pathway in LPS-induced RAW264.7 cells.

**Figure 6 foods-11-02488-f006:**
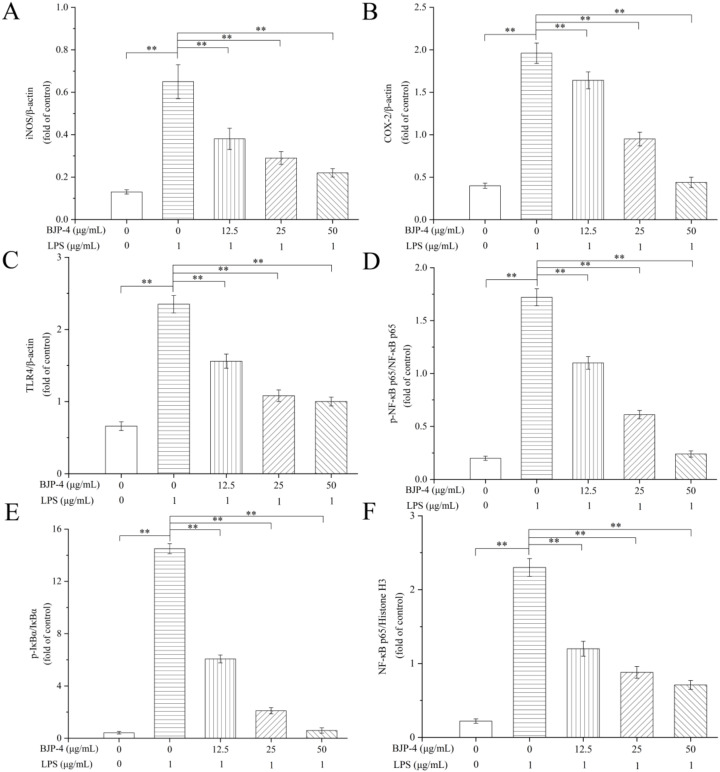
The protein levels of: (**A**) iNOS, (**B**) COX-2, (**C**) TLR4, (**D**) p-NF-κB p65/NF-κB p65, (**E**) p-IκBα/IκBα, (**F**) nuclear NF-κB p65. Values are expressed as mean ± SD. ** *p* < 0.01.

**Table 1 foods-11-02488-t001:** Primer sequences of RT-PCR analysis.

Gene			Product Size (bp)
TNF-α	F	TCTCATTCCTGCTTGTGG	195
	R	CTTGGTGGTTTGCTACG	
IL-6	F	TAACAGATAAGCTGGAGTC	115
	R	TAGGTTTGCCGAGTAGA	
IL-1β	F	AGCATCCAGCTTCAAATC	252
	R	ATCTCGGAGCCTGTAGTG	
iNOS	F	CCTGAGGGCTTTACTACAC	136
	R	GGTCTTTGCTGGCTGAT	
COX-2	F	GGGGTGATGAGCAACTA	199
	R	GGGTGCCAGTGATAGAG	
β-actin	F	CTGTGCCCATCTACGAGGGCTAT	155
	R	TTTGATGTCACGCACGATTTCC	

F, forward; R, reverse.

## Data Availability

Data is contained within the article.

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
