# Peer review of "An Acidic Polysaccharide with Anti-Inflammatory Effects from Blackened Jujube: Conformation and Rheological Properties"

_foods, 2022, doi:10.3390/foods11162488_

Round 1

Reviewer 1 Report

The manuscript titled “An acidic polysaccharide with anti-inflammatory effects from blackened jujube: Conformation and rheological properties” described the possible conformation and rheological properties of blackened jujube, including its therapeutic significance, which could be useful for food as well as pharmaceutical industries. The article has been written well, however needs minor revision, here are some remarks that have to be consider in further submission

·         Line no 77-83; Authors are suggested to write last paragraph of the introduction section with more clarity on the objectivity of the study. I felt that somehow clarity in the objectivity was missing. Kindly add some more specific information to fill the gap.  This will improve the over-all understanding of the readers.  

·         Methodologies are written nicely, however wherever reference required kindly add.

·         Results section- some of the images needed high resolution, kindly check with journal guidelines. In addition, figure no 5 is recommended to divide into two separate images, so gel images could have better visibility compare to current stages.  

·         The discussion of the results is partial add more details, more specific to section 3.4.1.

·         At few places, there is spacing issue, authors are suggested to correct it.

·         Some of the biological names needs to be written in italic. Kindly correct it throughout the manuscript.

·         Some of the comments has been marked in the pdf file, kindly refer to pdf pages

Author Response

Comment 1: Line no 77-83; Authors are suggested to write last paragraph of the introduction section with more clarity on the objectivity of the study. I felt that somehow clarity in the objectivity was missing. Kindly add some more specific information to fill the gap. This will improve the over-all understanding of the readers.

Response: Thank you very much for pointing this out, we have added and explained the objectivity of the study in the last paragraph of the introduction

Comment 2: Methodologies are written nicely, however wherever reference required kindly add.

Response: Thank you very much for your suggestion, we have added references in the methods section as appropriate.

Comment 3: Results section- some of the images needed high resolution, kindly check with journal guidelines. In addition, figure no 5 is recommended to divide into two separate images, so gel images could have better visibility compare to current stages.

Response: Thank you for your careful review, all image resolutions in the article have been modified according to journal guidelines, and in addition, we have divided Figure 5 into two separate images (Figure 5 and Figure 6).

Comment 4: The discussion of the results is partial add more details, more specific to section 3.4.1.

Response: Thank you for underlining this deficiency, the section 3.4.1 has been refined.

Comment 5: At few places, there is spacing issue, authors are suggested to correct it.

Response: It has been corrected.

Comment 6: Some of the biological names needs to be written in italic. Kindly correct it throughout the manuscript.

Response: We have rechecked the biological names throughout the manuscript and the formatting has been corrected.

Reviewer 2 Report

Dear Editor

Manuscript has been reviewed now and I found following points need to be addressed for the consideration of the manuscript

a. Mechanism of extraction process should be discussed in the introduction section

b. Sample preparation methods should be discussed properly

c. Figures are hardly to understand, therefore kindly improve the quality of figures

d. Conclusion should include future recommendations about the polysaccharide 

Author Response

Comment 1: Mechanism of extraction process should be discussed in the introduction section.

Response: Thank you for underlining this deficiency, we have added the extraction process of polysaccharides in the introduction section.

Comment 2: Sample preparation methods should be discussed properly.

Response: Thank you for pointing this out, we have described the sample preparation method in the introduction section and added references.

Comment 3: Figures are hardly to understand, therefore kindly improve the quality of figures.

Response: Thanks for your careful review, we have improved the quality of figures.

Comment 4: Conclusion should include future recommendations about the polysaccharide.

Response: Thank you for underlining this deficiency, the outlook for polysaccharides has been added at the conclusion.

Reviewer 3 Report

The paper investigated both the conformational and rheological properties of blackened jujube polysaccharide as well as its anti-inflammatory activities.

It is well structured. However, minor revision is needed.

In details:

- a deep english revision because many sentence had no significance (see for example lines 40-41 or 43-44 or 88-90, or ...)

- it is not clear the origin of the polysaccharide fraction. It should better explain

- 2.2. Section should be improved with the addition of more details for each type of analysis/measurement. In addition, before using an acronym, the corresponding words have to be reported.

- figure 1 has to be improved because the resolution is poor. Probably, it could be due to the very small single image

- figure 2: the same consideration performed for figure 1

Author Response

Comment 1: a deep english revision because many sentence had no significance (see for example lines 40-41 or 43-44 or 88-90, or ...).

Response: Thank you for underlining this deficiency, the sentences have been removed (lines 40-41 or 43-44 or 88-90).

Comment 2: it is not clear the origin of the polysaccharide fraction. It should better explain

Response: Thank you for pointing this out, the method of preparation of polysaccharide samples has been added in the introduction.

Comment 3: 2.2. Section should be improved with the addition of more details for each type of analysis/measurement. In addition, before using an acronym, the corresponding words have to be reported.

Response: Thank you for your careful review, more details of the experimental methods and references have been added in section 2.2. The corresponding word acronyms have been previously reported in the introduction section.

Comment 4: figure 1 has to be improved because the resolution is poor. Probably, it could be due to the very small single image.

Response: Thanks to your professional inspection, we reworked all the images in the manuscript and this resolution has been improved.

Comment 5: figure 2: the same consideration performed for figure 1

Response: Thanks for pointing this out, we reworked all the images in the manuscript and this resolution has been improved.

Round 2

Reviewer 2 Report

Dear authors

Kindly check grammatical errors throughout the manuscript and minimize them 

Author Response

Thank you for your careful review. We are very sorry for the mistakes in this manuscript and inconvenience they caused in your reading. The manuscript for all grammatical errors has been thoroughly revised and rewritten by us, and we hope it can meet the journal’s standard.